# Widespread contribution of transposable elements to the rewiring of mammalian 3D genomes

Mayank N. K. Choudhary [1,2,3], Kara Quaid[1,2,3], Xiaoyun Xing [1,2], Heather Schmidt[1,2] & Ting Wang [1,2] ✉

Transposable elements (TEs) are major contributors of genetic material in mammalian genomes. These often include binding sites for architectural proteins, including the multifarious master protein, CTCF, which shapes the 3D genome by creating loops, domains, compartment borders, and RNA-DNA interactions. These play a role in the compact packaging of DNA and have the potential to facilitate regulatory function. In this study, we explore the widespread contribution of TEs to mammalian 3D genomes by quantifying the extent to which they give rise to loops and domain border differences across various cell types and species using several 3D genome mapping technologies. We show that specific families and subfamilies of TEs have contributed to lineage-specific 3D chromatin structures across mammalian species. In many cases, these loops may facilitate sustained interaction between distant cis-regulatory elements and target genes, and domains may segregate chromatin state to impact gene expression in a lineage-specific manner. An experimental validation of our analytical findings using CRISPR-Cas9 to delete a candidate TE resulted in disruption of species-specific 3D chromatin structure. Taken together, we comprehensively quantify and selectively validate our finding that TEs contribute to shaping 3D genome organization and may, in some cases, impact gene regulation during the course of mammalian evolution.

Transposable elements (TEs) are major occupants of mammalian genetic real estate, including ~50% of the human genome[1,2]. TEs provide fodder to regulatory innovation by containing motifs that are very similar to transcription factor binding sites (TFBS) that can create functional TFBS upon mutations[3,4]. Consequently, these motifs have had a profound impact on remodeling gene regulatory networks[5–13] in human and other mammalian genomes. When these TE-derived binding sites belong to architectural proteins such as CTCF, ZNF143, etc., they can lead to certain TE families dramatically shaping genome folding[13–18]. Deleting TEs that supply CTCF binding sites involved in conserved 3D genome folding, can lead to the falling apart of these preserved higher-order chromosomal structures, underscoring their

functional importance[15]. Apart from carrying TF binding sites, elevated TE transcription can also contribute to dynamic 3D genome folding during development as has been shown in human pluripotent stem cells[19] and in early mouse embryos[20]. Similarly, the MIR family of ancient repeats also plays a critical role in 3D genome folding independent of CTCF[21]. They show a distinct local chromatin environment poised to recruit PolII, various transcriptional complexes, chromatin modifying enzymes, and histone modifications consistent with transcription and enhancers, some of which in part may help establish domain boundaries[21,22].

CTCF is known to play an important role in setting up the 3D genome architecture inside the nucleus of the cell[23,24]. Amongst its

[1]Center for Genome Sciences & Systems Biology, Washington University in St. Louis, St. Louis, MO 63110, USA. [2]Department of Genetics, Washington University in St. Louis, St. Louis, MO 63110, USA. [3]These authors contributed equally: Mayank N. K. Choudhary, Kara Quaid. ✉e-mail: twang@wustl.edu

many functions includes anchoring chromatin loops and insulating TAD boundaries in close concert with cohesin and other insulator proteins[25,26]. Studies profiling multiple species have underpinned CTCF evolution as one of the robust mechanisms of 3D genome remodeling; with differential CTCF binding leading to altered 3D genome folding[15,27]. Specifically, CTCF binding has evolved under two distinct forces of evolution and selection at play: a selective constraint to maintain existing domain boundaries and loop anchors, and a stronger push to evolve novel local interactions like intra-domain chromatin loops. With the latter fostering regulatory crosstalk and potentially rewiring regulatory networks, thereby pushing regulatory innovation and permitting phenotypic evolution. Thus, one regime maintains tight conservation of 3D genome folding, and the second regime pushes for divergence[14,15,28]. Combined with the waves of TE expansion, we are served a perfect recipe for sequence evolution, altered CTCF binding, 3D genome reorganization, and large-scale genome evolution.

In this study, we reconcile these two regimes of repeat-fueled CTCF evolution in the context of an evolving 3D genome. We hypothesize that TEs have been a rich source of sequence for the assembly and tinkering of higher-order chromosomal structures. To that end, we profile TE contributions at the class, family, and subfamily levels to higher-order chromosomal structures between humans and mice across multiple cell-types, as well as samples from dog and rhesus macaque, with the aim of understanding its impact on 3D genome evolution over deep time. We compare 3D genome folding patterns between species, with the aim to further our understanding of the role of TEs in the emergence and reshaping of genome topology throughout mammalian evolution.

## Results

### 8–37% of loop anchor and TAD boundary CTCF sites are derived from REs

In this study, we characterized 20 prior studies and collected 3D genome datasets across 52 cell types from 4 mammalian species (Fig. 1A). The profiled species (human, mouse, dog, and rhesus macaque) diverged 30–96 million years ago[29]. In general, we see that the proportion of the genome annotated as TEs increases from mouse to human, with dog having a slightly lower proportion than mice. The various classes of TEs also contribute differently (Fig. 1B). We started out by examining the contribution of all repetitive elements (REs), including transposable elements and genomic repeats, to loop anchor and TAD boundary CTCF sites, using genome-wide chromosome conformation capture-based datasets such as in situ Hi-C, HiChIP for CTCF and H3K27ac (associated with the higher activation of transcription and defined as an active enhancer mark), and ChIA-PET for H3K4me2 (enriched in cis-regulatory regions, in particular promoters, of transcriptionally active genes as well as genes primed for future expression during development in higher eukaryotes[30]). In studies where loop annotations and TAD boundary calls exist, we ran HiCCUPS (see Methods) to identify the underlying CTCF site that may be responsible for anchoring the loop or insulating the TAD boundary. For human cell types that lacked a corresponding CTCF ChIP data set, we used GM12878 ChIP data as a surrogate to call for CTCF occupancy. CH12-LX CTCF ChIP data was used as a surrogate for mouse cell types, which is supported by evidence of common CTCF binding across cell types in refs [31,32].

In our data collection, 8–37% of loop anchors and TAD boundary CTCF sites were derived from REs across the four species (Fig. 1C). We find that REs contribute to 8–12% of loop anchors and to 9–15% TAD boundary CTCF sites in a variety of human cell lines. In mouse cells, 15–22% of the loop anchors and 27–37% of TAD boundary CTCF sites were supplied by REs. The differences do not seem to be related to dataset resolution and are lower than random expectation (Supplementary Figs. 1, 2). The consistency of RE contribution to these CTCF sites underscores the pervasive nature of repeats in shaping 3D genome folding. Moreover, 16% of the CTCF loop anchors in rhesus macaque fibroblasts and as much as 14% of the TAD boundary CTCF sites in dog liver cells are derived from REs, highlighting that the commonality of this phenomenon is beyond human and mouse. Interestingly, REs have contributed more to CTCF loop anchors in rhesus macaque fibroblasts than human samples, and the TAD boundary contribution in the dog liver sample also looks more similar to what is observed across included human cell types. This may reflect the phylogenetic ancestry of these species as rhesus macaques and humans are more closely related, than mice, and dogs diverged earlier

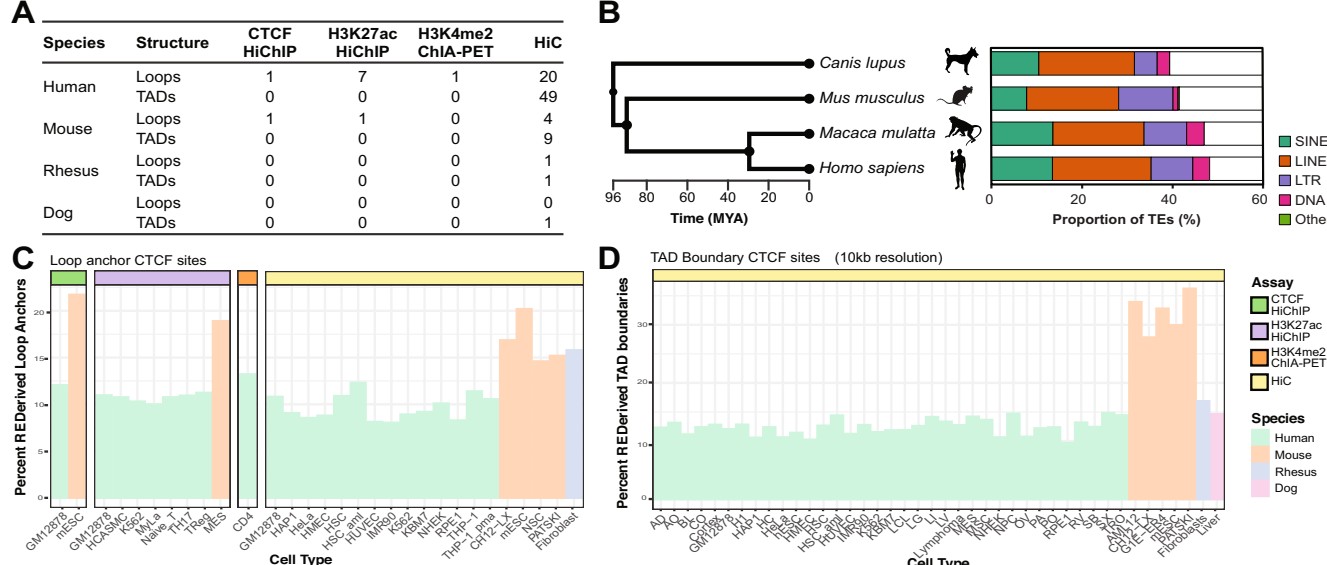

**Fig. 1 | Contribution of repeats to loop anchor and TAD boundary CTCF sites across 4 species. A** Catalog of publicly available datasets used as part of this study. **B** Phylogenetic tree of the four mammals (left). Contribution of TEs by class to total genome size (right). **C** Bar charts representing percentage of loop anchor and **D** TAD boundary CTCF sites derived from REs in a variety of human, mouse, rhesus macaque, and dog cell types.

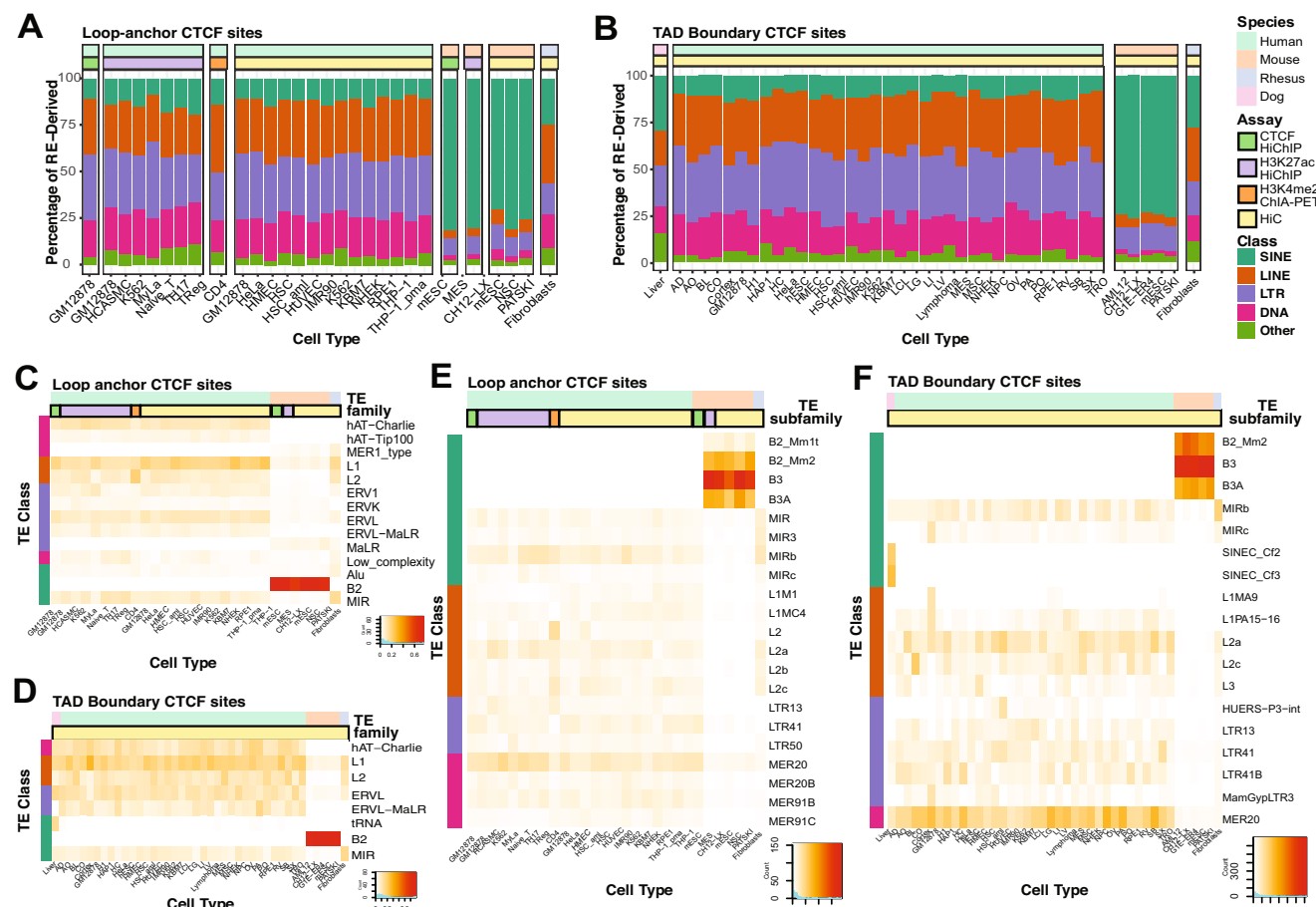

**Fig. 2 | Contribution of repeats class, family, and subfamily to loop anchor and TAD boundary CTCF sites. A** Stacked bar plots showing the distribution of RE-derived loop anchor CTCF binding site across major RE classes in the various cell types. **B** Stacked bar plots showing the distribution of RE-derived TAD boundary CTCF binding site across major RE classes in the various cell types. **C** Contribution of TE families to TE-derived loop anchor CTCF sites and **D** TE-derived TAD Boundary CTCF sites. **E** Contribution of TE subfamilies to TE-derived loop anchor CTCF sites and **F** TE-derived TAD Boundary CTCF sites.

and evolved independently (Fig. 1B). Based on the included data, TEs contribute to 8%-23% of loop anchor CTCF sites and 9-37% of TAD boundary CTCF sites across cell types and species.

## Lineage-specific contribution of repeat classes and families to loop anchor and TAD boundary CTCF sites

In the 4 species profiled, RE-derived loop anchor and TAD boundary CTCF sites were mostly (>90%) derived from TEs. Moreover, the class of origin of these TEs (SINE, LINE, LTR, DNA) also showed a species-biased distribution (Fig. 2A, B), as reported before[15]. In mouse, SINEs contribute the most to CTCF sites involved in genome folding even though they occupy ~5% lesser genomic space when compared to the three other species. This underscores the profound impact SINEs have had on shaping 3D genome folding in the mouse lineage. On the other hand, the human lineage had consistent contribution from LINE, LTR and DNA class of TEs. SINEs, however, did not contribute as much to either loop anchor or TAD boundary CTCF sites in the human lineage. TE class distribution did not seem to depend on cell type and/ or assay performed on the sample. Interestingly, we observe that the dog and rhesus macaque samples have a distribution of TE classes that lies between mouse and humans in terms of the relative contribution of all the four classes of TEs, with SINEs contributing a significant fraction in these species—higher than in human, but lower than mouse. On average, LINEs contribute to 4.6–37% of TE-derived loop anchors and TAD borders, SINEs contribute to 6.7–81.1%, LTRs contribute 8.9%-41%, and DNA transposons contribute to 1.8%-28% (Fig. 2A, B).

Next, we explored the contribution of TE families to TE-derived loop anchor and TAD boundary CTCF sites. The hAT-Charlie and hAT-Tip100 families of DNA transposons consistently contribute 1.6–15.8% of TE-derived loop anchor CTCF sites in human samples, but none in mouse or rhesus macaque as these families do not exist in those species. Instead, we see MER1_type of DNA transposons contributing to mouse and rhesus macaque loops and TAD boundaries, but not in humans. Amongst the LINEs, we see L1 contributing 3.9–25.4% and L2 family contributing 0.1–22.1% to TE-derived loop anchor CTCF sites. We see a variety of LTR families like ERV1, ERVK, ERVL, ERVL-MaLR (human-specific), and MaLR (not present in humans) contributing modestly (0.2%-12.8%) to TE-derived loop anchors, as has been reported before[15,17]. In the SINEs, the ancient MIR family contributes ubiquitously (0.7–16.9%) to TE-derived loop anchor CTCF sites in all three species. Lastly, the prolific B2 family of SINEs contributes extensively to loop anchors in mouse, accounting for 63.3–76.9% of the TE-derived loop anchor CTCF sites (Fig. 2C).

Next, we studied the TE family contribution to TAD boundary CTCF sites and observed similar overall contributions as in the case of loop anchor CTCF sites. We see a widespread contribution of TE-derived TAD boundary sites from the hAT-Charlie family of DNA transposons, L1 and L2 families of LINEs, ERV1, ERVL, and ERVL-MaLR families of LTRs, and B2 and MIR families of SINEs (0.2–70.1%) in the same species-biased manner, as observed before. In dogs, a dog-specific lysine-transfer RNA (tRNA)-derived SINE family of TEs contributes 29.3% of the TE-derived TAD boundary CTCF sites (Fig. 2D).

Next, we compared and contrasted the contribution of TE sub-families to TE-derived loop anchor and TAD boundary CTCF sites. In concordance with previously reported results, we saw enriched contributions from MER20, MER91B, MIRb, and L2a elements in humans, as well as B3 SINEs in mice[15,16]. Specifically, we looked at TE subfamily contribution in rhesus macaque and dog as these species have not been profiled before in direct comparison to human and mouse datasets. We observed that like humans, rhesus macaques also showed enrichment for MIR, MIR3, MIRb (highly enriched in humans) family of SINEs, as well as L2a (highly enriched in humans), L2b, L2c. However, unlike humans certain DNA transposons like MER20 and MER91B subfamily of elements were not enriched in rhesus macaque, and while LTR41 and LTR13 subfamily of LTR elements were enriched in humans, they were not found to contribute significantly to 3D genome folding in rhesus macaque (Fig. 2E). In dog, we saw that tRNA-derived canine-specific SINEC_Cf subfamilies of TEs were enriched for TE-derived domain boundary elements (Fig. 2F). While this has been reported before in the context of CTCF binding[14] its role in 3D genome folding was unknown. Together, these observations reflect lineage-specific differences in TE subfamily contribution to CTCF sites involved in 3D genome folding.

## TE-mediated CTCF binding site expansion as a mechanism of 3D genome folding

To study the effect of TE transposition on the evolution of 3D genome folding, we employed a previously published strategy[15] to annotate loops on two parameters - the origin of their CTCF site (TE-derived or non-TE derived) and their orthology status (conserved or lineage-specific) in matched cell-types, including liver samples from dog and human, and lymphoblastoid lines from mouse and human. TAD boundary orthology between the dog liver data from ref. [27] and the human liver data from ref. [33] were directly assessed. In the human liver data, 29.4% of the CTCF TAD boundaries were orthologous to the canFam3 reference genome, 14.2% of which were RE-derived. In contrast, 16.2% of human TAD boundaries that were non-orthologous to dog were RE-derived (Supplementary Fig. 3). 15.1% of the CTCF TAD boundaries in the dog liver data were orthologous to hg19 regions, 8.6% of which were RE-derived. Dog TAD boundaries that were non-orthologous to human were 13.9% RE-derived. Hence, non-orthologous structures were more likely to be RE-derived than orthologous structures. The overlap in CTCF binding between dogs and humans has been reported[14], but the contribution to 3D genome folding was previously unknown.

For functional validation, we focused on the lymphoblastoid lines in mice and human. Using in situ Hi-C data from the lymphoblastoid lineage, we compared annotated loop calls in GM12878 (human) and CH12-LX (mouse). Comparing CH12-LX to GM12878, we see that 49.3% (1643/3331) of the loops are mouse-specific. 18% (296/1643) of these mouse-specific loops were anchored at TE-derived CTCF sites (Fig. 3A). From here on out, we approached functional validation in human GM12878 cells, as it has the highest resolution in situ Hi-C data helping accurately assign functional loop anchors. We identified 1058 (14%) human-specific loops in GM12878 cells with at least one loop anchor CTCF exclusively derived from a TE. These loops could serve one of many purposes including but not limited to—enhancer-promoter loops and repressive loops. Using our loop function attribution strategy (detailed in Methods), we could attribute potential regulatory function to 410 of the 1058 (39%) TE-derived human-specific GM12878 loops (Fig. 3B). A similar fraction of TE-derived mouse-specific loops in CH12-LX (32.1%, 95 out of 296 loops) were speculatively functional (Fig. 3A). TE-derived human-specific enhancer-promoter loops were further divided based on strong/weak enhancer and active/weak promoter, as defined by ChromHMM annotations from ENCODE[34,35]. We found that 52.5% (148/282) of these loops potentially brought a strong enhancer near an active promoter.

Next, we sought to understand the differential contribution of TEs to conserved vs lineage-specific novel loops. We found that the lineage-specific B3A, B3, B2_Mm2 and B2_Mm1t subfamilies of SINEs contributed to mouse-specific loops in addition to their known role in maintaining conserved loops as reported before[15] (Fig. 3C). We also found that the MER20 family of DNA transposons contributes to both conserved loops as well as human- and mouse-specific loops.

To experimentally validate if these lineage-specific loops anchored at TE-derived CTCF sites play a regulatory role, we employed a candidate filtering and selection strategy (Fig. 3D, see Methods). Our top three candidate loops were anchored at an L1MC1, a MER57A1, and an LTR13 element. These TEs provide a CTCF binding site that anchors a loop predicted to recruit strong distal enhancers (supported by GeneHancer[36], HACER[37]) to promoters of highly expressed genes in blood cells. We further characterized one such candidate loop derived from an L1MC1 element. While we see the clear presence of a CTCF ChIP-seq peak as well as a loop anchor (as evidenced by a focal enrichment on the contact map, Fig. 3E, bottom) in GM12878 human cells, we observe that the syntenic region in mouse CH12-LX cells lacks a CTCF ChIP peak as well as chromatin loop (Fig. 3E, top). A genome browser view[38] of the candidate loop reveals that the left (5′) anchor lies close to the NCAPG2 promoter and the right (3′) anchor brings a ChromHMM-annotated (as well as HACER and an eQTL database cataloged) enhancer containing a regulatory SNP rs55752599 about 280 kb away in close proximity to each other (Fig. 3F). Interestingly, this loop is also anchored by a CTCF binding site derived from an MLT1H1 LTR element at the 5′ anchor end. Zooming in to the right 3′ end anchor (Fig. 3F, inset), we observe a MER52C LTR element possibly functioning as an enhancer (as evidenced by the p300 and H3K27ac ChIP peaks) that the L1MC1-supplied CTCF binding site potentially causes to loop over to NCAPG2, inducing a novel interaction and regulatory rewiring of the loci. In the syntenic region in mouse, we observe that these distal cis-regulatory elements also bear enhancer marks in CH12-LX cells, but they do not regulate the same target genes as in humans due to the absence of the L1MC1 TE and consequent chromatin looping.

Similarly, in the MER57A1- and LTR13-derived loop (see Supplementary Figs. 4-5), while the syntenic distal cis-regulatory elements bear enhancer marks in mouse CH12-LX cells, they are not wired to regulate the same target genes as in humans due to absence of the TE, the TE-derived CTCF motif, and consequent looping.

## L1MC1 TE anchors a human-specific, enhancer-promoter loop

Next, using CRISPR-Cas9 deletion, we validated the functionality of the L1MC1 element in anchoring the novel, human-specific chromatin loop (Supplementary Fig. 6). Upon deletion of L1MC1, the novel chromosomal structure collapsed as evidenced by the loss of focal enrichment in the homozygous TE knockout (KO) contact map (Fig. 4A, right) in comparison to the wild-type (WT) contact map (Fig. 4A, left). Inset shows a zoomed-in view of the focal enrichment at the loop anchors in the WT and the KO contact maps (Fig. 4A, inset).

Interestingly, the L1MC1-derived CTCF site also plays the role of a TAD domain boundary in GM12878, with active, expressed genes on one side and inactive, repressed genes on the other. Disrupting the domain border could allow regulatory elements to interact with these genes differently, changing the expression patterns seen in WT GM12878 cells. In the WT contact map, we see that 33% of the long-range (>30 kb) interactions within the region of interest (from the start of the affected domain to the end of the chromosome) take place upstream of the L1MC1 element, which is reduced to 24% in the KO contact map. This signifies the loss in domain-contained intra-TAD interactions upon deletion of the boundary element derived from L1MC1. Additionally, we see 28% of the long-range (>30 kb) inter-TAD interactions between the candidate domain (whose right 3′ boundary element is derived from the L1MC1 TE) and the downstream domain,

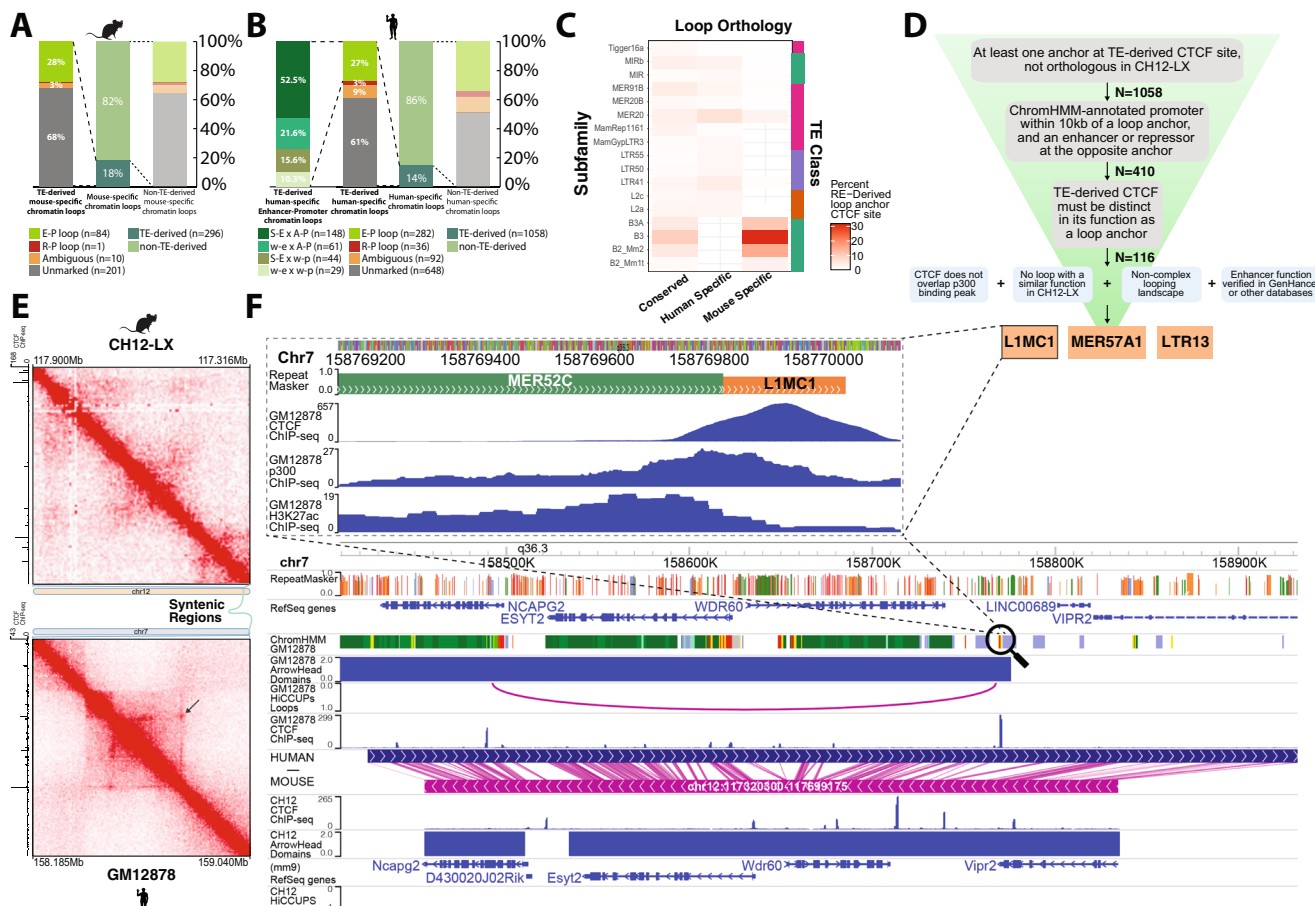

**Fig. 3 | TE-derived CTCF binding site expansion as a mechanism for novel 3D genome folding. A** Stacked bar plots showing the distribution of different classes of mouse-specific loops based on the origins of the CTCF loop anchor, i.e., TE-derived and non-TE-derived. **B** Stacked bar plots showing the distribution of different classes of human-specific loops based on the origins of the CTCF loop anchor, i.e., TE-derived and non-TE-derived. TE-derived, human-specific enhancer-promoter loops were further functionally annotated based on the strength of their enhancers and promoters. **C** TE subfamily contribution to conserved, human-specific, and mouse-specific loops in the lymphoblastoid cell lineage. **D** Schematic describing the methodology used to select a candidate loop and TE to validate. **E** in situ Hi-C contact map of a region containing L1MC1-derived CTCF site that forms a human-specific loop (arrow pointing) in GM12878 (bottom). The corresponding syntenic region in mouse in CH12-LX. **F** Genome browser screenshot displaying the L1MC1 candidate and the local chromatin and 3D landscape in human (main top) and mouse (main bottom). Zoomed in view of the L1MC1 candidate and the MER52C enhancer with CTCF, p300, and H3K27ac binding landscape (inset).

increasing from the 19% observed in the WT contact map. This again signifies the increase in "interaction leakage" that takes place after deleting the domain boundary element, which in this case was derived from the L1MC1 TE.

Thus, the L1MC1 element is necessary for anchoring a novel, human-specific loop and domain boundary in GM12878 cells.

To identify any changes in gene expression, RNA-seq was conducted on GM12878 wildtype cells and two independent knockout lines. The expression of genes within the disrupted domain was consistently, but not significantly, lower in the knockout lines (Fig. 4B, C). To interrogate possible mechanisms of genetic compensation, we used ATAC-seq to study the open chromatin landscape in wildtype GM12878 cells as well as the knockout GM12878 lines. Comparing the open chromatin landscape in wildtype and knockout GM12878 cells, we made the following interesting observations. The enhancer directly adjacent to the L1MC1-derived CTCF was much less accessible in the knockout GM12878 cells. This suggests that the enhancer activity is linked to the 3D conformation as it is lost upon deletion of the TE and the contained CTCF binding motif. Hence, the chromatin loop formation potentially leads to enhancer recruitment and the resulting open chromatin landscape.

While there was no notable change in chromatin accessibility within the disrupted TAD, the inactive downstream region gained ATAC peaks in the knockout GM12878 cells, which did not exist in that region in the wildtype GM12878 cells (Fig. 4D). ATAC peaks within the edited TAD showed decreased accessibility in the two knockout lines, while ATAC activity increased downstream of the edited TAD boundary, however, the magnitude of these changes was not outside of stochastic changes in chromatin accessibility seen genome-wide (Fig. 4E). Two downstream ChromHMM annotated enhancers that were not accessible in wildtype GM12878 cells have ATAC peaks called by MACS2 in the knockout lines (Fig. 4D). These peaks were also found to be annotated as ChromHMM enhancers in 1 out of 5 ENCODE GM12878 wild-type samples[35]. Indicating that these regions may play enhancer-like roles in either a subpopulation of GM12878 cells or other cell types. These newly accessible putative enhancers were cross-referenced using SCREEN[35]. A virtual 4C analysis was conducted anchoring on the two newly accessible peaks in the inactive domain, revealing no detectable increase in interaction between the newly-accessible enhancers and the upstream genes when comparing the mutant GM12878 lines to the wild-type GM12878 (See Supplementary Fig. 7).

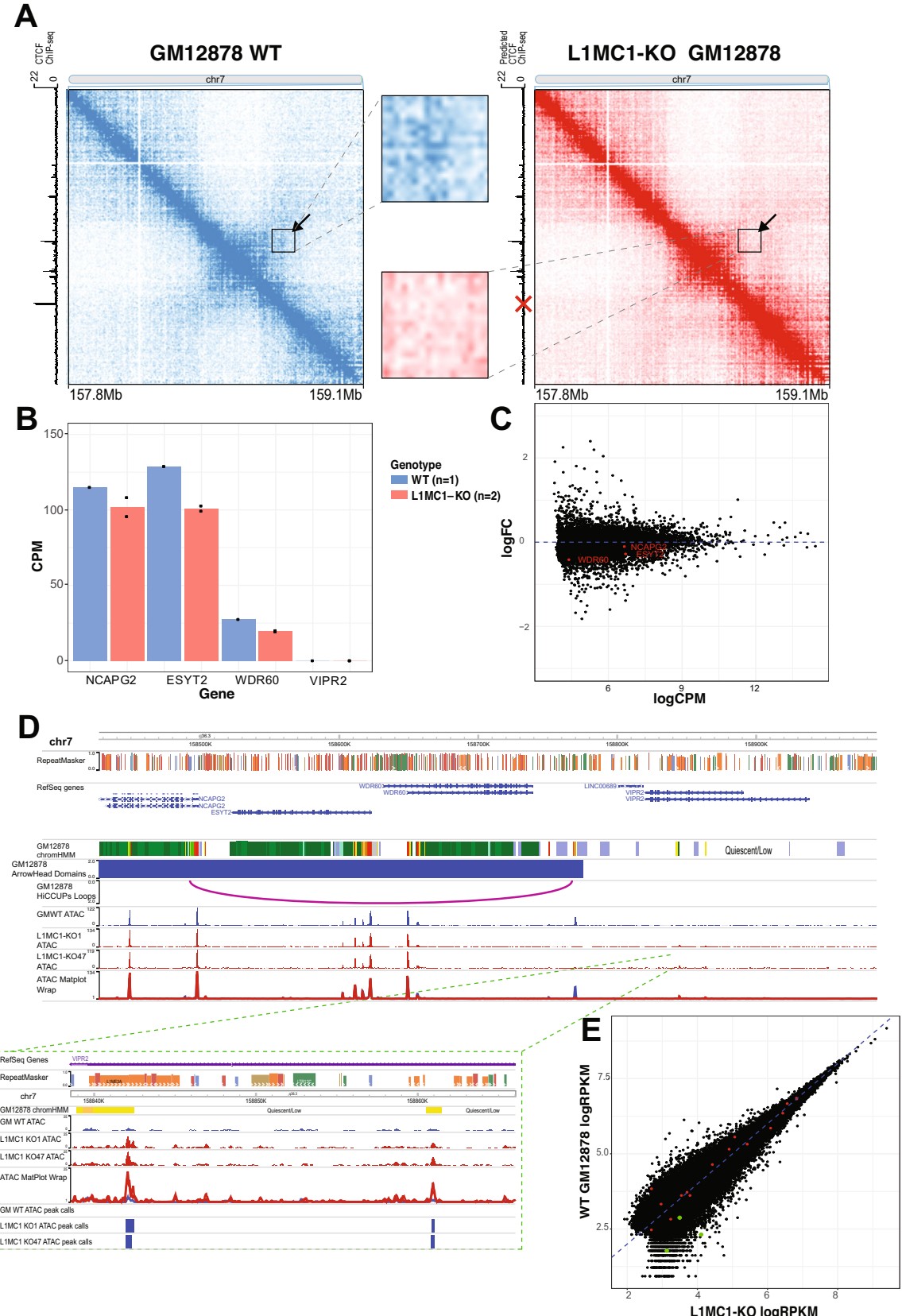

## Discussion

Divergence in gene regulation underpins the phenotypic diversity and evolutionary differences observed across species[39,40]. In part, this is mediated via evolution of cis-regulatory elements and higher-order chromosomal structures that function in tandem to link the former to their target genes. While studies over the last two decades have revealed that a large portion of mammalian cis-regulatory elements is derived from TEs[11,12], their role in the evolution of higher-order

**Fig. 4 | L1MC1-anchored novel human loop is lost upon CRISPR-Cas9 deletion of TE. A** Contact map generated using GM12878 Hi-C[2] data overlaid with arrow and a black square box around the novel, human-specific loop (left). The contact map generated using Hi-C[2] data in L1MC1-KO GM12878 with arrow and a black square box around where the novel, human-specific loop would have showed focal enrichment (right). Both contact maps have been KR-normalized. Publicly available CTCF ChIP-seq data was used for determining predicted CTCF binding. Zoomed in view of the focal enrichment characteristic of the loop (inset). Red X marks the L1MC1 loci deleted. **B** Bar chart of gene expression (exonic read CPM) per gene in the target region. Expression from L1MC1-KO lines was averaged in the bars.

Individual data points are represented by dots. **C** Scatter plot of gene expression fold change (logFC denotes log$_2$Fold Change (knockout/wildtype)) and absolute gene expression (log$_2$CPM of exonic reads) of all genes. **D** WashU Epigenome Browser screenshot of open chromatin landscape near the deleted L1MC1-CTCF. Inlay shows downstream ATAC peaks in the inactive region of wildtype GM12878 cells. **E** Scatter plot of normalized tag density for ATAC-seq data (log$_2$RPKM within ATAC peaks). Peaks shown had an RPKM of at least 2 in all samples. Green points are ATAC peaks present in the mutants in the "inactive" region downstream of the deleted L1MC1-CTCF. Red points are ATAC peaks in the "active" domain upstream of the deleted L1MC1-CTCF.

chromosomal structure has been unexplored until recently. Moreover, TE-derived cis-regulatory elements often function in a cell-type and species/clade-specific manner. In fact, the candidate we validate in this study is an example of both these features.

We had earlier shown that TEs contributed a substantial number of loop anchors in mouse and human 3D genomes, a fraction of which were co-opted to help maintain conserved higher-order chromosomal structures[15]. In this study, we further explored in detail the contribution of TEs to novel, lineage-specific 3D genome structures like loop anchors and TAD boundaries. Like cis-regulatory elements, we observe that TE-derived higher-order chromosomal structures are often cell-type and lineage-specific.

In addition, TE transposition provides an attractive mechanism to deposit novel CTCF binding sites and give rise to new 3D genome structures be it loops or TADs, accelerating cis-regulatory evolution. On the one hand, the finding that in most cases TEs contribute less to CTCF-bound TAD boundaries and loop anchors than random expectation may imply an overall deleterious relationship between TEs and these important 3D structures. On the other hand, TEs should be completely absent if they played no functional role. Their common presence within these functional regions implies in many cases they are not deleterious to these structures and have played a role in shaping them. The CTCF motifs within these TEs combined with their proliferative presence in loop anchors and domain boundaries suggest that these TEs are not only not detrimental to genomic function in this context, but may provide genomic function. Moreover, not all TE families/ subfamilies are equal. Some subfamilies like B2/ B3 SINEs in mice and SINEC_Cf in dogs are clearly enriched (over random expectation) and have actively contributed to chromatin organization of the mouse and dog genome. Once TEs disperse CTCF binding sites, selection can exploit the availability of these functional CTCF binding sites in the neighborhood to assemble novel regulatory interactions and form networks. Such a mechanism was recently speculated to play a role in the formation of super-interacting promoter groups by the ERVL-MaLR family of TEs that supply ZNF143 binding sites aiding corticogenesis in the human fetal brain[17].

Deletion of a candidate L1MC1 TE, that anchored a regulatory SNP-containing enhancer (which was coincidentally TE-derived as well) end of a human-specific enhancer-promoter loop, led to the collapse of the novel chromatin structure - showing the necessity of the TE-derived CTCF site, and consequently the TE. Hence, TEs have played a key role in establishing novel higher-order chromosomal structures and may be components of lineage-specific gene regulatory networks. The deletion of the TE-derived CTCF site coincided with a small, but consistent decrease in gene expression within the disrupted TAD. There were marked changes in chromatin accessibility at putative enhancers in the downstream, previously inactive region. These newly accessible putative enhancers may underlie the lack of ablation or even significant decrease in gene expression. The lack of significant change also demonstrates the plasticity and robustness that exists in gene regulation and consequent gene expression. The gain in accessibility of these downstream enhancers may also be due to increased interactions with regions within the disrupted active TAD. The overall effect on chromatin and gene expression we see is very small, and it is not

uncommon to see no changes in expression after the deletion of a CTCF motif[40–45]. Our observation of a very small change may be due to the deletion only affecting the genome conformation in a small fraction of the cells, making the overall effect size smaller at the level of an entire cell population.

Thus, we provide evidence of how bursts of lineage-specific TE activity can provide CTCF binding sites resulting in lineage-specific higher-order chromosomal structures that rewire local gene regulation, pushing regulatory innovation, ultimately permitting the emergence of novel evolutionary phenotypes. Taken together, our observations and results represent the regulatory innovation that TEs bring to host genomes.

By analyzing publicly available high-resolution Hi-C datasets from four different species, we aimed to identify, quantify, and validate TE transposition as a widespread phenomenon that contributes to lineage-specific genome folding patterns and consequently lineage-specific gene regulation across multiple mammalian species. We found that 8–37% of loop anchor and TAD boundary CTCF sites were derived from repeats in the 37 cell types that were profiled. We outline lineage-specific contribution of TE classes, families, and subfamilies to loop anchor and TAD boundary CTCF sites, highlighting TE families like MIR (common), tRNA (dog), and B2 (mouse) SINEs, L1 (common) and L2 (all but mouse) LINEs, ERV-family (common) of LTRs and hAT-Charlie (humans) family of DNA transposons that have contributed extensively to CTCF sites involved in 3D genome folding. Further, we characterize lineage-specific chromatin loops, showing that a significant fraction (14–18%) of these are anchored at TE-derived CTCF binding sites. A substantial fraction (27–28%) of these TE-derived, lineage-specific loops were predicted to be enhancer-promoter loops, representing the regulatory innovation TEs unintentionally bring along as they infect and invade the host genome. We also find that some of these lineage-specific TE-derived loops contribute to divergent gene expression due to lineage-specific transcriptional regulation. Finally, we used CRISPR-Cas9 to delete a candidate TE (L1MC1) that was predicted to anchor an enhancer-promoter loop as well as a TAD domain boundary in human lymphoblastoid cell lines. Upon deletion of the element, we see collapse of the novel loop as well as the domain boundary, underscoring the importance of TEs in the context of 3D genome remodeling and regulatory innovation. Taken together, our findings shed light on the extent to which TEs have contributed to loop anchor and TAD boundary CTCF sites in mammalian species, consequently rewiring the 3D genome and accelerating regulatory crosstalk and innovation.

## Methods

### Dataset GEO accession numbers
Hi-C data for GM12878 and CH12-LX were obtained from ref. [46]. Loop calls from H3K27me ChIA-PET experiments on CD4 + T-cells were obtained from refs. [25,47]. Loop calls from H3K27ac Hi-ChIP experiments on GM12878, HCASMC, K562, MyLa, Naive T-cells, TH17, TReg, and MESC samples were obtained from ref. [48]. Loop calls from CTCF Hi-ChIP experiments on GM12878 and mESC samples were obtained from ref. [49]. Loop calls from in-situ Hi-C experiments on THC-1 samples were

obtained from ref. [50]. Loop calls from in-situ Hi-C experiments on HAP1 cells were obtained from ref. [46]. Loop and TAD calls from in-situ Hi-C experiments on HSC samples were obtained from ref. [51]. Loop calls from additional Hi-C experiments on HSC samples were obtained from ref. [52]. Loop and TAD calls from AML12 Hepatocytes were obtained from[53]. TAD calls from in-situ Hi-C experiments on hESCs were obtained from ref. [54]. TAD calls from in-situ Hi-C experiments on Cortex, hESC, and IMR90 cells were obtained from ref. [25]. TAD calls from in-situ Hi-C experiments on Lymphoma samples were obtained from ref. [55]. TAD calls for LCL samples were obtained from ref. [56]. TAD calls from adrenal gland, aorta, bladder, cortex, GM12878, hESC, hippocampus, IMR90, lung, liver, left ventricle, mesoderm, mesenchymal stem cell, NPC, ovary, pancreas, Psoas muscle, right ventricle, small bowel, spleen, and trophoblast-like cells were obtained from ref. [33]. TAD calls on RPE1 cells were obtained from ref. [57]. Links and cell-type abbreviations for the datasets are available in Supplementary Table 1.

## Loop anchor and TAD boundary CTCF−RE intersection

We used published HiCCUPS output[26] from all of the above-mentioned cell and tissue types to generate a list of unique loop anchors. We used MotifFinder[26] in conjunction with CTCF ChIP-seq data (GEO-accession: GSM935611)[58] to identify loops that were anchored by CTCF sites. We used BEDTools[59] to overlap loop anchor CTCF motifs with RepeatMasker[60] (RMSK v4.0.7, for hg19, mm9, canFam3, and rheMac2) with a required 10 bp overlap of the CTCF motif with a repetitive element (RE) to classify RE-derived loop anchor CTCF sites. Loops with at least one identified RE-derived anchor CTCF (RE-derived loop) or two anchors with no RE-derived CTCF sites (non-RE derived loop) were considered for analysis of RE-derived loop counts.

## TAD boundary CTCF−RE intersection

Similarly, we included published TAD boundaries for analysis. We considered the region 5 kb upstream and 5 kb downstream of the published TAD borders as the boundaries for analysis to standardize the resolution between studies. We used the same MotifFinder analysis and BEDTools intersections as the loop analysis to identify CTCF-derived TAD boundaries. TADs with at least one identified RE-derived CTCF site were classified as RE-derived TADs, and TADs with two anchors with no RE-derived CTCF site were classified as non-RE-derived TADs to be considered for summary analysis. For analysis of RE class contribution to CTCF TAD boundaries, samples with the same cell type, species, and assay were considered together with average class contributions (sum of unique RE-derived CTCF loop anchors/sum of unique CTCF loop anchors).

## TE class and family distribution

We used RepeatMasker v4.0.7[60] with its slow search parameter to obtain a comprehensive list of REs as well as their class, family, and subfamily in the hg19, mm9, rheMac2, and canFam3 genome assemblies. We used the RE counts (described above) to quantify their contribution to loop anchor and TAD boundary CTCF sites.

## Loop orthology check

We converted CH12-LX loop calls in the mm9 assembly to hg19 coordinates using liftOver[61] using a sequence match rate of 0.1. 3245 out of 3331 mouse peaks were successfully lifted over from mm9 to hg19 to enable us to compare peak annotations across species. Mouse loop anchor pairs that fell within a minimum threshold (half loop length, or 50 kb) of an existing GM12878 loop anchor pair were considered to be conserved in humans. The 50 kb threshold is intended to account for errors in cross-species liftOver.

## TAD orthology check

We identified CTCF TAD boundaries in human and dog livers by identifying all CTCF motifs within the 40 kb TAD boundary, and intersecting those with human and dog liver CTCF ChIP peaks from ref. [14]. We converted the human liver CTCF TAD boundaries in the hg19 assembly to canFam3 using liftOver[61] with a sequence match rate of 0.1. 1146 out of 1216 CTCF TAD boundaries were successfully lifted from hg19 to canFam3. Lifted TAD boundary CTCF motifs that fell within 50 kb of an existing dog liver CTCF TAD boundary were considered to be conserved.

## Loop annotation strategy

To annotate loops with putative regulatory function, we used ChromHMM state annotations in the respective cell type. We considered if there was a ChromHMM-annotated promoter (TssA, TssAFlnk) within 10 kb of one loop anchor and either a ChromHMM-annotated enhancer (Enh, EnhG, (and EnhA1, EnhA2 for humans)) or polycomb repression (ReprPC, ReprPCWk) at the opposite loop anchor. ChromHMM chromatin states are as described in ref. [62] with "unmarked" indicating genomic sequences without evidence of regulatory information.

## Candidate selection and filtering

Candidate selection and filtering were conducted using loop calls on GM12878 and CH12-LX cells from ref. [26]. To be considered for CRISPR validation, we required that a GM12878 loop had at least one anchor at a RE-derived CTCF that is not orthologous in CH12-LX cells. Then we looked for loops that had a promoter within 10 kb of one anchor and an enhancer or repressor within 10 kb the opposite anchor, as annotated by ChromHMM[58]. We also required that the TE-derived CTCF was not nearby another active CTCF binding site. CTCF ChIP data was downloaded[58], and MACS2 was used to call peaks to determine locations with active CTCF binding[63]. Loop anchors with more than one active CTCF site within 15 kb were excluded. Furthermore, the RE-derived CTCF could not overlap a p300 binding peak[58] to avoid interrupting enhancer function. Enhancer-promoter interactions that had been validated in the GeneHancer database were preferred, but not required.

## Cell culture methods

GM12878 cell lines were cultured according to ENCODE standards. Cells were cultured in 10 mL of RPMI1640 media (Gibco, 1187-085) with 100 U/ml penicillin−streptomycin (Gibco, 15140-122) and 15% fetal bovine serum (Corning, 35-011-CV). They were grown at a density of 200K−800K cells/mL in T-25 flasks and incubated at 37 °C with 95% $CO_2$.

## CRISPR-Cas9 mediated genome engineering

We used a CRISPR workflow to delete the L1MC1-derived CTCF loop anchor (identified through the outlined steps above) from cultured GM12878 cells. The pU6-(BbsI)_CBh-Cas9-T2A-BFP plasmid (Addgene, 64323) and pU6-(BbsI)_CBh-Cas9-T2A-mCherry plasmid (Addgene, 64324) were used as CRISPR delivery vectors. We referenced CRISPOR[64] and CRISPRScan[65] to design several pairs of sgRNAs with minimal off-target effects and high cutting efficiency. We used three sgRNA oligos; one upstream of the L1MC1 CTCF (sequence: ggtgataatgacgttactgtgggg), and two downstream of the L1MC1 CTCF (sequences: ctctgggggggtcagg tatgt and gacaggcagaaggtcacgcctgg). These sgRNA oligos were designed with compatible overhangs to be cloned into BbSI digested BFP and mCherry CRISPR vectors. We constructed BFP-CRISPR vectors that expressed the sgRNA upstream of the candidate TE and mCherry-CRISPR vectors that express the downstream sgRNA. The BFP and mCherry-CRISPR vectors were co-transfected into GM12878 cells. 24 h post-transfection, we used flow cytometry (Sony SH800 Cell Sorter) to single-cell sort BFP and mCherry double-positive cells into 96-well plates for clonal expansion for 28 days. The targeted region of nine clones was

amplified through PCR and Sanger-sequenced to verify the CRISPR deletion. four had heterozygous deletions and five had homozygous deletions.

## Hi-C experiments

The Hi-C datasets used to validate the L1MC1-derived loop were generated using the standardized protocol from the 4DN consortium[66]. Specifically, 5 million GM12878 WT and L1MC1-KO cells were pelleted at $300 \times g$. They were cross-linked using 1% formaldehyde for 10 min and quenched with 2.5 M glycine (final concentration 0.2 M) and incubated on a rocker at room temperature for 5 min. The cells were centrifuged at $300 \times g$ at 4 °C, and pellets were flash-frozen in dry ice and ethanol. Cells were lysed using 250 ul of ice-cold Hi-C lysis buffer (10 mM Tris-HCL pH8.0, 10 mM NaCl) and incubated on ice for 15 min. The cells were pelleted at $2500 \times g$ and washed with ice-cold lysis buffer. The pellets were resuspended in 0.5% sodium dodecyl sulfate (SDS) ad incubated at 62 °C for 10 min. 145 μl of water and 25 μl of 10% Triton X-100 (Sigma, 93443) were added, mixed, and incubated at 37 °C for 15 min. The DNA was digested using 100 U of MboI in 10X NEBuffer 2 overnight at 37 °C with rotation. MboI was inactivated at 62 °C for 20 min. Then at room temperature, the DNA overhangs are filled with dNTPs and biotinylated using biotin-14-dATP (Life Technologies, 19524-016) and 40U of DNA Polymerase I, Large (Klenow) Fragment (NEB, M0210). The mixture is incubated at 37 °C with rotation for 45 min. The ends were then ligated using 120 μl of 10X NEB T4 DNA ligase buffer (NEB, B0202), 100 μl of 10% Triton X-100, 12 μl of 10 mg/ml Bovine Serum Albumin (100X BSA), and 5 μl of 400 U/μl T4 DNA Ligase (NEB, M0202) and incubated at room temperature for 4 h. Nuclei were pelleted at $2500 \times g$ and resuspended in 10 mM Tris-HCl, pH8; 0.5 M NaCl; 1% SDS solution, and treated with 20 mg/ml proteinase K (NEB, P8102) to be incubated for 30 min at 55 °C, then overnight at 68 °C. The pellet is cleaned using 2× volumes of pure ethanol and incubating at −80 °C for 2 h. Next, the blunt-end fragments were resuspended in 130 μl of 1X Tris buffer (10 mM Tris-HCl, pH 8). And sheared to 400–700 bp fragments using six cycles (30 s on, 30 s off) on Bioruptor® Pico sonication device (Diagenode Cat# B01060010). 75 μl of 10 mg/ml Dynabeads MyOne Streptavidin T1 beads (Life Technologies, 65602) were used to pull down the biotinylated fragments, from which an Illumina library was built and amplified through 10–12 rounds of PCR. These fragments were finally analyzed using shallow paired-end sequencing (1.4M-4M reads) to ensure library quality (complexity, contact statistics) before Hi-C$^2$. Libraries that passed the quality check were used as pools for Hi-C$^2$ experiments.

We generated two in situ Hi-C libraries of GM12878 cells with verified L1MC1 deletions and 1 Hi-C library of wildtype GM12878 cells. Details about all in-situ Hi-C libraries are available in Supplementary Tables 2–4. Hi-C data was processed as described in ref. [26]. Hi-C data were visualized using the web version of JuiceBox[67].

## Hi-C$^2$ probe design

We followed the approach used in ref. [46] to design probes for Hybrid Capture Hi-C (Hi-C$^2$). Briefly, we designed probe sequences to a target region (chr7:157.1-159.1MB), which surrounded the L1MC1-derived CTCF loop anchor we had removed with CRISPR. These sequences were required to be near MboI restriction sites. We executed a three-pass probe design strategy, incrementally increasing several parameters, including the distance of the probe from the MboI restriction site, GC content, probe density in gaps, and the number of repetitive bases. Probes with identical sequences and overlapping probes were removed from the pool. Using this strategy, we identified 3195 probes for the region surrounding the L1MC1-derived CTCF loop anchor. We appended primer sequences (15 bp) to both ends of the 120 bp probe sequence. Similarly to ref. [46], we conducted probe construction and hybrid selection with our study-specific sequences. Hi-C$^2$ libraries were sequenced to a depth of 100 M reads.

## Hi-C$^2$ analysis

Hi-C$^2$ data were aligned and processed using Juicer[68]. Matrices for analysis were generated using juicer dump (parameters: oe, NONE, BP 5000). Percentages of long-range (>30 kb) interactions were calculated within the region of interest (chr7:158400000-159000000). The intra-TAD interactions upstream of the L1MC1 element were within chr7: 158490000-158770000, and the inter-TAD interactions were between chr7:158490000-158770000 and chr7:158770000-159100000.

## RNA-seq experiments

500k cells from each of the genetically edited clones and GM12878 WT cells were treated with Trizol for lysing. RNA was extracted using a Zymo RNA Clean & Concentrator-5 kit. 200 ng of RNA from each sample was used to generate RNA-seq libraries using the NuGEN Universal Plus mRNA-Seq + UDI kit. The libraries underwent 2 × 75 paired end sequencing and the resulting data were aligned to hg19 using STAR[69]. Gene-wise read counts were extracted using HTSeq2[70] and CPM was calculated for GAPDH, ACTB, NCAPG2, ESYT2, WDR60, and VIPR2 using edgeR[71].

## ATAC-seq experiments

ATAC-seq was performed as described in ref. [72]. 100K–200K cells from the WT GM12878 and homozygous knockout lines were collected on day 6 of repriming. Cells were resuspended and centrifuged at $500 \times g$ for 5 min at 4 °C. The supernatant was aspirated and cells were washed once with cold PBS (0.04% BSA). The cells were resuspended in 300 μl DNaseI solution (20 mM Tris pH 7.4, 150 mM NaCl, 1 × reaction buffer with MgCl2, 0.1 U/μl DNaseI) on ice for 5 min. Subsequently, 1 ml of PBS (0.04% BSA) was added, and the mixture was centrifuged at $500 \times g$ for 5 min at 4 °C. The cells were washed twice in PBS + BSA and resuspended in 100 μl ATAC-seq RSB (10 mM Tris pH 7.4, 10 mM NaCl, 3 mM MgCl$_2$ in water) with 0.1% NP40, 0.1% Tween-20, and 0.01% Digitonin and incubated on ice for 3 min to lyse. After lysis, 1 mL of ATAC-seq RSB with 0.1% Tween-20 was added and mixed by inversion. Next, the nuclei were collected by centrifugation at $500 \times g$ for 5 min at 4 °C. The nuclei were resuspended in 20 μL 2 × TD buffer (20 mM Tris pH 7.6, 10 mM MgCl$_2$, 20% Dimethyl Formamide), and 50,000 nuclei were transferred to a tube with 2 × TD buffer filled up to 25 μL. 25 μL of transposition mix [2.5 μL Transposase (100 nM final) (Illumina, 20034197), 16.5 μL PBS, 0.5 μL 1% digitonin, 0.5 μL 10% Tween-20, and 5 μL water] was added. The transposition reactions were mixed and incubated for 30 min at 37 °C with gentle tapping every 10 min. Reactions were purified with the Zymo DNA Clean and Concentrator-5 kit (Zymo Research, D4014). The ATAC-seq library was amplified for nine cycles on a PCR machine. The PCR reaction was purified with Sera-Mag Select beads (Cytiva, 29343057) using double size selection with 27.5 μL beads (0.55 × sample volume) and 50 μL beads (1.55 × sample volume). The ATAC-seq libraries were quantitated by Qubit assays and sequenced by an Illumina NextSeq platform. QC and analysis on ATAC-seq libraries were performed using AIAP[73]. Peaks in the ATAC data were called using MACS2[63] and visualized on the WashU Epigenome Browser[38]. Differentially accessible peaks between the wildtype and knockout cells were identified using DiffBind[74].

## Reporting summary

Further information on research design is available in the Nature Portfolio Reporting Summary linked to this article.

## Data availability

The data that support this study are available from the corresponding author upon reasonable request. Hi-C$^2$, RNA-seq, and ATAC-seq data

generated in this study have been deposited in the Gene Expression Omnibus (GEO) database under accession code GSE222526. Publicly available sequencing datasets analyzed in this study include loop data that can be found in the supplementary information of[51]. H3K4me2 ChIA-PET can be accessed from GSE32677. H3K27ac HiChIP data can be accessed from GSE101498. CTCF HiChIP data can be accessed from GSE80820. HiC data can be accessed from the Genome Sequence Archive under accession code CRA000108, Zenodo under accession code 1244182, and from GEO accession codes GSE35156, GSE65126, GSE71831, GSE95476, GSE87112, GSE96800, GSE95116, GSE76479, GSE128678, GSE63525, GSE74072, and GSE144126.

## Code availability

All custom scripts are available at https://github.com/twlab/novel3dte[75].

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

## Acknowledgements

We thank members of the Wang Lab for helpful discussions related to the project; Jessica Hoisington-López and Maria Lynn Jaeger from The Edison Family Center for Genome Sciences and Systems Biology for assistance with sequencing; Matthew Patana from the Siteman Flow Cytometry core for FACS expertise. This work was supported by NIH R01HG007175(T.W.), U01CA200060(T.W.), U24ES026699(T.W.), U01HG009391(T.W.), UM1HG011585(T.W.), and T32HG000045-18(K.Q.).

## Author contributions

M.N.K.C., K.Q., and T.W. conceived and designed this study; M.N.K.C. and K.Q. analyzed the data, performed the experiments, generated the sequencing libraries, and wrote the manuscript with inputs from T.W. X.X. and H.S. performed the ATAC-seq experiments and provided text and revision for the manuscript. All authors subsequently edited and approved the final manuscript.

## Competing interests

The authors declare no competing interests.
