## [Peer Review File · Nature Communications]

REVIEWER COMMENTS

Reviewer #1 (Remarks to the Author):

In this work, Choudhary and collaborators study the contribution of TEs in the formation of loop-anchor CTCF sites. To do so, the authors first analyze the proportion of TEs present in loop-anchor CTCF sites in human, macaque, mouse, dog genomes using publicly available HiC, HiChIP and ChIA-PET data. Then, focusing on two species (human and mouse), the authors define the number of TEs loop-anchor CTCF sites specific for each of these two species, selecting one candidate loop (L1MC1) for further functional analyses. These analyses included CRISPR-Cas9 deletion of this single locus, coupled with HiC, ATAC-seq and RNA-seq experiments to test whether depletion of binding site had an impact on 3D structure. From their studies the authors conclude that 'TEs contribute significantly to 3D genome organization and continuously shape it to affect gene regulation during the course of mammalian evolution over deep time'. Although the idea is interesting I find that the conclusions are premature based on the data presented. The functional analyses are based on the experimental validation of one single locus in a human cell line (GM12878, an EBV transformed B-lymphocyte). There are also some issues regarding the approach and methodology conducted that need clarification.

One of my concerns is related to the first section of the results '8-39% of loop anchor and TAD boundary CTCF sites are derived from Res' and it deals with the high heterogeneity of the data compared, which is heavily unbalanced among species. This can compromise the results and conclusions derived. The authors are analysing 4 species (human, macaque, mouse and dog) but are in fact comparing very different cell types and numbers: one set of HiC data derived from dog liver, one set of HiC data from fibroblasts in the case of macaque, then other cell lines in mouse and human. Cell types and numbers are not balanced among species. Have the author control against variability between cell types? In the case of the mouse, for example, the results from the PATSKI cell line are clearly outlier. Have the authors an explanation for this? As such, the statement 'This may reflect the phylogenetic ancestry of these species as rhesus macaques and humans are more closely related' might be premature. Comparing the same cell type across the species will provide more solid results. An alternative will be to reduce the comparative study to human and mouse cell lines, which in fact are the species studied in much detail in the subsequent sections. For example, in the following section 'Lineage-specific contribution of repeat classes and families to loop anchor and TAD boundary CTCF sites', the dog is not even included/mentioned (i.e., not present in panels 2A-E).

As for the third section 'TE-mediated CTCF binding site expansion as a mechanism of novel 3D genome folding', the methodology followed to select 3 candidate loop regions out of the 1,000 initial reported in the previous section should be described in more detail. There is a reference to the figure, but the approach is not detailed.

The fourth section 'L1MC1 TE anchors a novel human-specific, enhancer-promoter loop' needs some technical clarifications. First, the CRISPR-Cas9 results with the human cell line GM12878 are not explained in sufficient detail. Second, there are some issues with the HiC data that need clarification. In the M&M section it is said that 'we generated 3 in situ Hi-C libraries of GM12878 cells with verified L1MC1 deletions and 1 Hi-C library of wild type GM12878 cells', but Supplementary Tables 1 & 2 only refer to the statistics of two libraries. Is the control missing? This is not entirely clear based on the name of the library. There is also disparity between libraries' statistics: a high percentage (40-60%) of duplicates with a final number of valid pairs of 28 M for one library and 15M for the other one. This low level of resolution can compromise the results. Also, Figure 4A depicts a HiC map derived from GM12878 cells, but this map is extracted from data already publicly available. Authors

should show the HiC maps of their own experimental control. As for the ATAC-seq, number of replicates are not specified, nor is comparison among peaks genome-wide provided.

Additional comments:

Line 45: Species can be named here.

Line 55: Not convinced the analyses provide, overall, insights into 'gene regulation during the course of mammalian evolution over deep time'. The in-deep analyses are done comparing human and mouse cell lines and functionally testing one single locus in a human cell line.

Lines 61-63: Sounds repetitive given the previous sentence.

Lines 96-97: The statement 'multiple species and cell-types' is somehow misleading as only human and mouse data is derived from different cell types.

Lines 107-108: Please, specify the difference between TEs and Res in this context.

Lines 119-120: Needs reference to figure.

Lines 128-129: This statement can be compromised by the fact that different cell types are compared between species.

Line 198: Not clear the term 'base species' in this context.

Lines 206-207: Which are the parameters for the definition of strong/weak enhancer (promoter)? Described the methodology more into detail.

Lines 210-213: Reference to Figure 3C?

Lines 217-220: This is not part of the results, reads more like a conclusion and it underlines one of issues of the study: low number of species compared with equivalent data /cell types available.

Figure 3A: Not clear what is the meaning of 'unmarked' in this context.

Reviewer #2 (Remarks to the Author):

Choudhary et al. have performed a comparative analysis of the contribution of TEs to 3D mammalian genome organisation. They identify TE families that overlap with loop anchors or TAD borders in a species-specific manner, and demonstrate how the deletion of one such element in human cells leads to 3D genome rearrangement.

This is a well conducted study, with clearly described epigenomic analyses and a nice example where the role of TEs as 3D genome organisers is genetically validated. The only dampener on my excitement is the fact that many of the findings had already been described by the same group (PMID: 31973766), using partially overlapping data and similar methodology. The authors acknowledge this throughout the manuscript, but the fact remains that the new aspects of this work struggle to come through. So my first suggestion to the authors would be to spend more time describing new findings, such as the comparison with macaque and dog (the dog-specific SINE family intrigued me) or perhaps any potential tissue-specific differences (given a much larger array of cell lines were analysed now). In addition, I have the following queries/suggestions:

1. GM12878 CTCF data was used as a surrogate for cell types without ChIP data. What is the estimate error associated with this approach? I.e., how variable are CTCF peaks across cell types for which there is ChIP data?

2. To what extent are the percentages for TE intersections with loops/TADs affected by differences in data quality/quantity? For example, do PATSKI cells really have nearly double the contribution of TEs to loop-anchor CTCF sites than the rest of the mouse cell

types (Fig. 1C), or are the data biased in some way?

3. It would be useful to compare the identified overlaps with a random control, such as shuffled loop anchors / TAD boundaries. This would highlight enriched TE classes/families (even though I appreciate some are already quite obvious), and in case data quality did affect loop/TAD calling significantly, mitigate against these differences.

4. On page 9: "Using our loop function attribution strategy (detailed in Methods), we could attribute a regulatory function to 410 of the 1058 (39%) TE-derived human-specific GM12878 loops (Fig. 3B)". Being mindful of the endless debate about the meaning of the word 'function', I would reconsider its use in this sentence, since it is based simply on chromHMM annotations.

5. In supplementary figure 3 it would be useful to add some genomic context (features, not just coordinates) for reference, to know where peaks would have been expected to emerge in the KOs.

Miguel Branco

Responses to referees for revised manuscript NCOMMS-21-49591, "Widespread contribution of transposable elements to the rewiring of mammalian 3D genomes and gene regulation" by Choudhary and Quaid et al.

We thank the referees and everyone involved in the editorial process for their time and effort in the consideration of our manuscript, and for their contribution to its improvement. In response to their thoughtful suggestions, we have performed additional lab work and analyses to support the conclusions of the study, address important details, and improve readability. We believe that this revised manuscript satisfactorily addresses all referee concerns, and we detail the specific revisions made in response to the original reviews throughout the remainder of this document.

Referee #1 (Remarks to the Author):

In this work, Choudhary and collaborators study the contribution of TEs in the formation of loop-anchor CTCF sites. To do so, the authors first analyze the proportion of TEs present in loop-anchor CTCF sites in human, macaque, mouse, dog genomes using publicly available HiC, HiChIP and ChIA-PET data. Then, focusing on two species (human and mouse), the authors define the number of TEs loop-anchor CTCF sites specific for each of these two species, selecting one candidate loop (L1MC1) for further functional analyses. These analyses included CRISPR-Cas9 deletion of this single locus, coupled with HiC, ATAC-seq and RNA-seq experiments to test whether depletion of binding site had an impact on 3D structure. From their studies the authors conclude that 'TEs contribute significantly to 3D genome organization and continuously shape it to affect gene regulation during the course of mammalian evolution over deep time'. Although the idea is interesting I find that the conclusions are premature based on the data presented. The functional analyses are based on the experimental validation of one single locus in a human cell line (GM12878, an EBV transformed B-lymphocyte). There are also some issues regarding the approach and methodology conducted that need clarification.

We thank the referee for their attention to our manuscript and for bringing forward analyses and clarifications that were overlooked. In response to these excellent suggestions, we have performed additional analyses, which are discussed below and in the revised text.

Specific Comments

1a. One of my concerns is related to the first section of the results '8-39% of loop anchor and TAD boundary CTCF sites are derived from REs' and it deals with the high heterogeneity of the data compared, which is heavily unbalanced among species. This can compromise the results and conclusions derived.

1b. The authors are analysing 4 species (human, macaque, mouse and dog) but are in fact comparing very different cell types and numbers: one set of HiC data derived from dog liver, one set of HiC data from fibroblasts in the case of macaque, then other cell lines in mouse and human. Cell types and numbers are not balanced among species.

1c. Have the author control against variability between cell types? In the case of the mouse, for example, the results from the PATSKI cell line are clearly outlier. Have the authors an explanation for this? As such, the statement

1d. 'This may reflect the phylogenetic ancestry of these species as rhesus macaques and humans are more closely related' might be premature. Comparing the same cell type across the species will provide more solid results. An alternative will be to reduce the comparative study to human and mouse cell lines, which in fact are the species studied in much detail in the subsequent sections. For example, in the following section 'Lineage-specific contribution of repeat classes and families to loop anchor and TAD boundary CTCF sites', the dog is not even included/mentioned (i.e., not present in panels 2A-E).

(Response 1.1) We thank the referee for their insightful critique and recommendations. In response, we have conducted additional analyses to determine the influence of variation between included data sets, as well as increased the depth of analyses of the dog data to compare the same cell type across species. We have also adjusted the text to emphasize the balance of human and mouse data relative to rhesus and dog.

Specifically in response to suggestions 1a and 1c, we have compared the resolution of the included datasets by determining the number of loops called for each dataset and checking for any biases in terms of percentage of repetitive element (RE) derived loop anchors across our data set. This has been included as Supplementary Figure 1 and Response Figure 1B, below. It does not seem like there is any influence of variability of dataset resolution on the percentages. In terms of the PATSKI outlier, we are very grateful to reviewers for pointing this out. Looking further into the PATSKI dataset, we discovered an error in our processing pipeline for this dataset. We have now rectified the error and re-analyzed the data. All figures and text references have been updated to reflect this. To more closely investigate the outlier PATSKI dataset, we included an independent PATSKI data set. This dataset was downloaded in Hi-C format from the Disteche Lab in the 4DN database (<https://data.4dnucleome.org/files-processed/4DNFIPUDOX9Q/>), and called loops according to Rao et. al 2014. The original dataset was generated in the Aiden lab, part of a 2016 publication from Darrow and Huntley. The loops called from the PATSKI Hi-C data from the Disteche lab closely overlap the loops called by Darrow and Huntley (Response Figure 1A, below). With the updated pipeline, they also have a nearly identical contribution from repetitive elements (Response Figure 1B, below).

(Response Figure 1)

In response to reviewer point 1b, we agree with the reviewer that our findings are not comprehensive across all cell types and therefore should not be expressed as hard boundaries for the TE contribution to 3D genome organization across the entire species. We have adjusted the text to clarify that these results only apply to the cell types included.

In response to reviewer point 1d, we have conducted additional analyses. To expand our analysis and make better use of the dog data, we have analyzed TAD boundary orthology between the dog liver data from Rudan et. al and the human liver data from Schmitt et. al. As shown in Response Figure 2, in the human liver data, 29.4% of the CTCF TAD boundaries were orthologous to the canFam3 reference genome, with 14.2% RE-derived. 16.2% of human TAD boundaries that were non-orthologous to dog were RE-Derived. 15.1% of the CTCF TAD boundaries in the dog liver data were orthologous to hg19 regions, 8.6% of which were RE-derived. Dog TAD boundaries that were non-orthologous to human were 13.9% RE-Derived. This is in line with previous findings that non-orthologous structures were more likely to be RE-derived than orthologous structures (Choudhary et. al, 2020). This data is now included in the supplementary information as Supplementary Figure 3.

(Response Figure 2)

2. As for the third section 'TE-mediated CTCF binding site expansion as a mechanism of novel 3D genome folding', the methodology followed to select 3 candidate loop regions out of the 1,000 initial reported in the

previous section should be described in more detail. There is a reference to the figure, but the approach is not detailed.

(Response 1.2) We thank the reviewer for the suggestion. This methodology is detailed in the methods section, we have clarified the main text to make it more clear to readers.

3. The fourth section 'L1MC1 TE anchors a novel human-specific, enhancer-promoter loop' needs some technical clarifications. First, the CRISPR-Cas9 results with the human cell line GM12878 are not explained in sufficient detail. Second, there are some issues with the HiC data that need clarification. In the M&M section it is said that 'we generated 3 in situ Hi-C libraries of GM12878 cells with verified L1MC1 deletions and 1 Hi-C library of wild type GM12878 cells', but Supplementary Tables 1 & 2 only refer to the statistics of two libraries. Is the control missing? This is not entirely clear based on the name of the library. There is also disparity between libraries' statistics: a high percentage (40-60%) of duplicates with a final number of valid pairs of 28 M for one library and 15M for the other one. This low level of resolution can compromise the results. Also, Figure 4A depicts a HiC map derived from GM12878 cells, but this map is extracted from data already publicly available. Authors should show the HiC maps of their own experimental control. As for the ATAC-seq, number of replicates are not specified, nor is comparison among peaks genome-wide provided.

(Response 1.3) We thank the reviewer for this suggestion. We have revised the text to make it clear that there are two replicates of the L1MC1 knockout line. The details of the altered sequence of these clones are now included in the supplementary materials as Supplementary Figure 4.

With respect to the difference in resolution between the knockout HiC² libraries, the second knockout library is only used to verify the lack of loop anchor in cells with a deleted L1MC1-derived element. That library was not directly compared to any other library directly, so we feel that the difference in resolution is acceptable.

We also conducted HiC² on unedited, wildtype GM12878 cells to use as a more suitable control to compare with our edited cell lines. We have adjusted Figure 4 to reflect the wildtype data produced in our lab using the same HiC² protocol as was used in the knockout lines.

Additional comments:

Line 45: Species can be named here.

- We thank the reviewer for the suggestion, and the text now reads "in several mammalian species, including humans, mice, rhesus macaque, and dog".

Line 55: Not convinced the analyses provide, overall, insights into 'gene regulation during the course of mammalian evolution over deep time'. The in-deep analyses are done comparing human and mouse cell lines and functionally testing one single locus in a human cell line.

- We thank the reviewer for the suggestion, and the text now reads "in some cases, affect gene regulation". We have also removed "gene regulation" from the title of the manuscript to dampen expectations of functional claims.

Lines 61-63: Sounds repetitive given the previous sentence.

- We thank the reviewer for the suggestion, and the text now reads "TEs provide fodder to regulatory innovation by means of containing transcription factor motifs as well as motifs that are very similar to transcription factor binding sites (TFBS) that can create functional TFBS upon precise mutations[12,13]. Consequently, these motifs have had a profound impact on remodeling gene regulatory networks[3–11] in human and mammalian genomes at large."

Lines 96-97: The statement 'multiple species and cell-types' is somehow misleading as only human and mouse data is derived from different cell types.

- We thank the reviewer for the suggestion, and the text now reads "between humans and mice."

Lines 107-108: Please, specify the difference between TEs and Res in this context.

- We thank the reviewer for the suggestion, and the text now reads “repetitive elements (REs), including transposable elements and genomic repeats.”

Lines 119-120: Needs reference to figure.

- We thank the reviewer for the suggestion, and the figure is now referenced.

Lines 128-129: This statement can be compromised by the fact that different cell types are compared between species.

- We thank the reviewer for the suggestion, and the text now reads “similar to what is observed across included human cell types”

Line 198: Not clear the term ‘base species’ in this context.

- We thank the reviewer for the suggestion, and the text now reads “we approached functional validation in human GM12878 cells.”

Lines 206-207: Which are the parameters for the definition of strong/weak enhancer (promoter)? Described the methodology more into detail.

- We thank the reviewer for the suggestion, and the methods section now includes “ChromHMM chromatin states are as described in Ernst and Kellis, 2017, with “unmarked” indicating genomic sequences without evidence of regulatory information.”

Lines 210-213: Reference to Figure 3C?

- We thank the reviewer for the suggestion, and the figure is now referenced.

Lines 217-220: This is not part of the results, reads more like a conclusion and it underlines one of issues of the study: low number of species compared with equivalent data /cell types available.

- We thank the reviewer for the suggestion, and the erroneous section has been removed.

Figure 3A: Not clear what is the meaning of ‘unmarked’ in this context.

- We thank the reviewer for the suggestion, and the methods section now includes “ChromHMM chromatin states are as described in Ernst and Kellis, 2017, with “unmarked” indicating genomic sequences without evidence of regulatory information.”

Referee #2 (Remarks to the Author):

Choudhary et al. have performed a comparative analysis of the contribution of TEs to 3D mammalian genome organisation. They identify TE families that overlap with loop anchors or TAD borders in a species-specific manner, and demonstrate how the deletion of one such element in human cells leads to 3D genome rearrangement.

This is a well conducted study, with clearly described epigenomic analyses and a nice example where the role of TEs as 3D genome organisers is genetically validated. The only dampener on my excitement is the fact that many of the findings had already been described by the same group (PMID: 31973766), using partially overlapping data and similar methodology. The authors acknowledge this throughout the manuscript, but the fact remains that the new aspects of this work struggle to come through. So my first suggestion to the authors would be to spend more time describing new findings, such as the comparison with macaque and dog (the dog-specific SINE family intrigued me) or perhaps any potential tissue-specific differences (given a much larger array of cell lines were analysed now). In addition, I have the following queries/suggestions:

We thank the referee for their kind words and their insights, which we believe have made significant improvements to the manuscript. In response to their thoughtful suggestions, we have conducted additional analyses detailed below, and clarified the text of the manuscript.

Specific Comments:

1. GM12878 CTCF data was used as a surrogate for cell types without ChIP data. What is the estimate error associated with this approach? I.e., how variable are CTCF peaks across cell types for which there is ChIP data?

(Response 2.1) We thank the referee for this suggestion. In response we have conducted analysis comparing GM12878 CTCF ChIP peak calls (Rao et. Al, 2014) to CTCF ChIP Peak calls from the IMR90 cell line (ENCODE: ENCSR000EFI). We also investigated the use of GM12878 CTCF ChIP data as a proxy for loop anchor CTCF motif identification.

(Response Figure 3)

We found that 88.9% of CTCF ChIP peaks found in GM12878 cells were also found in IMR90 cells. The correlation coefficient of normalized CTCF ChIP reads within peaks between GM12878 and IMR90 was 0.73. There is also a precedent in the literature that CTCF binding is generally conserved among cell types (Azazi et. al 2020, Chen et. al, 2012).

Based on this evidence, we acknowledge the limitation, and suggest it is justifiable to use GM12878 ChIP data as a surrogate for cell types without ChIP data.

2. To what extent are the percentages for TE intersections with loops/TADs affected by differences in data quality/quantity? For example, do PATSKI cells really have nearly double the contribution of TEs to loop-anchor CTCF sites than the rest of the mouse cell types (Fig. 1C), or are the data biased in some way?

(Response 2.2) We thank the referee for their suggestion. In response we have included information about data quality and depth in the revised supplement (Supplemental Figure 1), which suggests that the data quality including the number of loop anchors or TAD boundaries called does not seem to affect the distribution of TE contribution.

With respect to the comment about the PATSKI cell line, we have investigated further into this cell line, by including a second, independent data set, detailed in the response to Reviewer 1 (Response 1.1).

3. It would be useful to compare the identified overlaps with a random control, such as shuffled loop anchors / TAD boundaries. This would highlight enriched TE classes/families (even though I appreciate some are already quite obvious), and in case data quality did affect loop/TAD calling significantly, mitigate against these differences.

(Response 2.3) We thank the referee for this suggestion. As shown in Response Figure 4, we have conducted additional analysis of a shuffled set of repetitive elements to find the distribution of RE-derived loop anchors and TAD boundaries where the location of repetitive elements is randomized. This analysis is included in supplemental materials (Supplementary Figure 2)

(Response Figure 4)

We found that with randomly located TEs, the percentage of CTCF loop anchors and TAD boundaries derived from REs evens out to approximate the genome wide RE rate, as expected, which is more than what we see with the real locations of the REs.

4. On page 9: “Using our loop function attribution strategy (detailed in Methods), we could attribute a regulatory function to 410 of the 1058 (39%) TE-derived human-specific GM12878 loops (Fig. 3B)”. Being mindful of the endless debate about the meaning of the word ‘function’, I would reconsider its use in this sentence, since it is based simply on chromHMM annotations.

(Response 2.4) We thank the referee for this suggestion. The text has been edited to emphasize that these loops only have “potential regulatory function” rather than defined regulatory function. We have also removed “gene regulation” from the title of the manuscript to dampen expectations of functional claims.

5. In supplementary figure 3 it would be useful to add some genomic context (features, not just coordinates) for reference, to know where peaks would have been expected to emerge in the Kos

(Response 2.5) We thank the referee for this suggestion. The figure has been edited to include the annotation of the TAD upstream of the potentially newly active enhancers, as well as the location of the edited L1MC1 element.

Citations

- Azazi, D., Mudge, J.M., Odom, D.T. et al. Functional signatures of evolutionarily young CTCF binding sites. *BMC Biol* 18, 132 (2020). <https://doi.org/10.1186/s12915-020-00863-8>
- Chen H, Tian Y, Shu W, Bo X, Wang S (2012) Comprehensive Identification and Annotation of Cell Type-Specific and Ubiquitous CTCF-Binding Sites in the Human Genome. *PLoS ONE* 7(7): e41374. <https://doi.org/10.1371/journal.pone.0041374>
- Choudhary, M.N., Friedman, R.Z., Wang, J.T. et al. Co-opted transposons help perpetuate conserved higher-order chromosomal structures. *Genome Biol* 21, 16 (2020). <https://doi.org/10.1186/s13059-019-1916-8>
- Dekker, J., Belmont, A., Guttman, M. et al. The 4D nucleome project. *Nature* 549, 219–226 (2017). <https://doi.org/10.1038/nature23884>
- Ernst J, Kellis M. Chromatin-state discovery and genome annotation with ChromHMM. *Nat Protoc.* 2017 Dec;12(12):2478-2492. doi: 10.1038/nprot.2017.124. Epub 2017 Nov 9. PMID: 29120462; PMCID: PMC5945550. <https://doi.org/10.1038/nprot.2017.124>
- Rao SS, Huntley MH, Durand NC, Stamenova EK, Bochkov ID, Robinson JT, Sanborn AL, Machol I, Omer AD, Lander ES, Aiden EL. A 3D map of the human genome at kilobase resolution reveals principles of chromatin looping. *Cell.* 2014 Dec 18;159(7):1665-80. doi: 10.1016/j.cell.2014.11.021. Epub 2014 Dec 11. Erratum in: *Cell.* 2015 Jul 30;162(3):687-8. PMID: 25497547; PMCID: PMC5635824.
- Vietri Rudan M, Barrington C, Henderson S, Ernst C, Odom DT, Tanay A, Hadjur S. Comparative Hi-C reveals that CTCF underlies evolution of chromosomal domain architecture. *Cell Rep.* 2015 Mar 3;10(8):1297-309. doi: 10.1016/j.celrep.2015.02.004. Epub 2015 Feb 26. PMID: 25732821; PMCID: PMC4542312.
- Schmidt D, Schwalie PC, Wilson MD, Ballester B, Gonçalves A, Kutter C, Brown GD, Marshall A, Flicek P, Odom DT. Waves of retrotransposon expansion remodel genome organization and CTCF binding in multiple mammalian lineages. *Cell.* 2012 Jan 20;148(1-2):335-48. doi: 10.1016/j.cell.2011.11.058. Epub 2012 Jan 12. Erratum in: *Cell.* 2012 Feb 17;148(4):832. PMID: 22244452; PMCID: PMC3368268.
- Zhang, J., Lee, D., Dhiman, V. et al. An integrative ENCODE resource for cancer genomics. *Nat Commun* 11, 3696 (2020). <https://doi.org/10.1038/s41467-020-14743-w>

REVIEWERS' COMMENTS

Reviewer #1 (Remarks to the Author):

I am satisfied with the revisions made by the authors.

Reviewer #2 (Remarks to the Author):

The authors have made improvements to their manuscript that address some of the technical concerns I had.

I just have one remaining discussion point, which the authors may choose to address/incorporate. It is about whether % RE overlap with CTCF-bound loops/TADs is sufficient to make the argument that REs have actively contributed to chromatin organisation, or whether an enrichment over random expectation is required to make this point.

When considering all REs, it is clear that they are depleted from loops/TADs (comparison with shuffled REs), yet some families/subfamilies (e.g., B2/B3 SINEs in mouse) are clearly enriched. Most data are displayed as % overlap, so it's hard to assess enrichment over random in a lot of cases.

But is enrichment required to make the point? One on hand, a lack of enrichment suggests a random process, whereby most REs just happen to be there (some select ones may still be functional). On the other, one could argue that if they played no functional role, they should be completely absent from loops/TADs, otherwise they may disrupt them. I think the fact that the presence of CTCF motifs was taken into account in the analysis helps with the latter argument. Still, I'm not sure there's a clear-cut answer, and so wondered whether the authors would like to express their view in the Discussion section of the manuscript.

Miguel Branco

Responses to referees for revised manuscript NCOMMS-21-49591, "Widespread contribution of transposable elements to the rewiring of mammalian 3D genomes and gene regulation" by Choudhary and Quaid et al.

We thank the referees and everyone involved in the editorial process for their time and effort in the consideration of our manuscript, and for their massive contribution to its improvement. In response to their thoughtful suggestions, we have updated the discussion section to include our thoughts on the contribution of TEs to 3D chromatin structures as compared to random chance.

Referee #1 (Remarks to the Author):

I am satisfied with the revisions made by the authors.

- We thank the Referee #1 for their incredibly helpful suggestions and critiques during the first revision, and are happy that they are satisfied with our revisions.

Referee #2 (Remarks to the Author):

The authors have made improvements to their manuscript that address some of the technical concerns I had.

I just have one remaining discussion point, which the authors may choose to address/incorporate. It is about whether % RE overlap with CTCF-bound loops/TADs is sufficient to make the argument that REs have actively contributed to chromatin organisation, or whether an enrichment over random expectation is required to make this point.

When considering all REs, it is clear that they are depleted from loops/TADs (comparison with shuffled REs), yet some families/subfamilies (e.g., B2/B3 SINEs in mouse) are clearly enriched. Most data are displayed as % overlap, so it's hard to assess enrichment over random in a lot of cases.

But is enrichment required to make the point? One on hand, a lack of enrichment suggests a random process, whereby most REs just happen to be there (some select ones may still be functional). On the other, one could argue that if they played no functional role, they should be completely absent from loops/TADs, otherwise they may disrupt them. I think the fact that the presence of CTCF motifs was taken into account in the analysis helps with the latter argument. Still, I'm not sure there's a clear-cut answer, and so wondered whether the authors would like to express their view in the Discussion section of the manuscript.

Miguel Branco

- We thank Miguel for his thoughtful contribution to the success of this paper during the first revision and are excited to include our thoughts on this important matter in the final version.
- The discussion section of the manuscript now includes, "The finding that in most cases, TEs contribute less to CTCF TAD bounds and loop anchors than the random expectation may imply an overall deleterious relationship between TEs and these important structures. However their common presence within these functional regions implies in many cases they are not deleterious to these structures. The CTCF motifs within these TEs combined with their proliferative presence in loop anchors and domain boundaries suggest that these TEs are not only not detrimental to genomic function in this context, but may provide genomic function."